# Semi-supervised Convolutional Networks for Translation Adaptation with Tiny Amount of In-domain Data

## Abstract

In this paper, we propose a method which uses semi-supervised convolutional neural networks (CNNs) to select in-domain training data for statistical machine translation. This approach is particularly effective when only tiny amounts of in-domain data are available. The in-domain data and randomly sampled general-domain data are used to train a data selection model with semi-supervised CNN, then this model computes domain relevance scores for all the sentences in the general-domain data set. The sentence pairs with top scores are selected to train the system. We carry out experiments on 4 language directions with three test domains. Compared with strong baseline systems trained with large amount of data, this method can improve the performance up to 3.1 BLEU. Its performances are significant better than three state-of-the-art language model based data selection methods. We also show that the in-domain data used to train the selection model could be as few as 100 sentences, which makes fine-grained topic-dependent translation adaptation possible.

## 1 Introduction

Statistical machine translation (SMT) systems are trained on bilingual parallel and monolingual data. The training corpora typically come from different sources, and vary across topics, genres, dialects, authors' written styles, etc., which are usually referred as "general domain" training data. Here the word "domain" is often used to indicate some combination of all above and other possible hidden factors (Chen et al., 2013). At run time, the content to be translated may come from a different domain. Due to the mismatch in "domains", it is possible to achieve better performance by adapting the SMT system to the test domain (in-domain).

However, manually creating training data to match the test domain is not a preferred solution, because 1) sometimes the test domain is not known when training the model, and it could change from sentence to sentence; 2) even if the test domain is pre-determined, the resources required and slow turnaround in data collection process will still delay the system development process.

Therefore, training data selection is widely used for domain adaptation in statistical machine translation (Zhao et al., 2004; Lü et al., 2007; Yasuda et al., 2008; Moore and Lewis, 2010; Axelrod et al., 2011; Duh et al., 2013; Axelrod et al., 2015). Data selection techniques select monolingual or bilingual data that are similar to the in-domain seed data based on some criteria, which are incorporated into the training data. The most successful data selection approaches (Moore and Lewis, 2010; Axelrod et al., 2011) train $n$-gram language models on in-domain text to select similar sentences from the large general-domain corpora according to the cross entropy. Furthermore, (Duh et al., 2013) obtained some gains by extending these approaches from $n$-gram models to recurrent neural network language models (Mikolov et al., 2010). To train the in-domain language model, a reasonable size in-domain data set, which typically includes several thousands of sentences, is required. In (Axelrod et al., 2011; Duh et al., 2013), the sizes of the in-domain data sets are 30K and over 100K sentences respectively.

However, we do not always have access to large or even medium amounts of in-domain data. With the growth of social media, new domains have emerged which need machine translation but

which have very limited in-domain data, maybe just a few hundred sentence pairs. What's more, if one wishes to build a large scale topic-specific MT system with hundreds of topics, it is prohibitively expensive to collect tens of thousands of in-domain sentences for each topic.

In this paper, we try to address this challenge, i.e., domain adaptation with very limited amounts of in-domain data. Inspired by the success of convolutional neural networks (CNNs) applied to image and text classification (Krizhevsky et al., 2012; Kim, 2014; Johnson and Zhang, 2015a; Johnson and Zhang, 2015b), we propose to use CNN to classify training sentence pairs as in-domain or out-of-domain sentences. To overcome the problem of limited in-domain data, we propose to augment the original model with semi-supervised convolutional neural networks for domain classification.

Convolutional neural networks (CNNs) (LeCun and Bengio, 1998) are feed-forward neural networks that exploit the internal structure of data through convolution layers; each computation unit processes a small region of the input data. CNN has been very successful on image classification. When applying it to text input, the convolution layers process small regions of a document, i.e., a sequence of sentences or words. CNN has been gaining attention, and is now used in many text classification tasks (Kalchbrenner et al., 2014; Zeng et al., 2014; Johnson and Zhang, 2015b; Yin and Schütze, 2015; Wang et al., 2015).

In many of these studies, the first layer of the network converts words to word embeddings using table lookup. The word embeddings are either trained as part of CNN training, or pre-trained (thus fixed during model training time) on an additional unlabled corpus. The later is termed semi-supervised CNN. Given tiny amounts of in-domain data, the information learned in these pre-trained word embeddings is very helpful.

We use a small amount of in-domain data, such as the development set, as the positive sample and randomly select the same number of sentences from the general-domain training data as the negative sample to form the training sample for training the CNN classification model. This is a typical supervised learning setting. To compensate the limit of in-domain data size, we use $word2vec$ (Mikolov et al., 2013) to learn the word embedding from a large amount of general-domain data.

Together with the labeled data, these word embeddings are fed to the convolution layer as additional input to train the final classification model. This is a semi-supervised framework. The learned models are then used to classify each sentence in the general-domain training data based on their domain relevance score. The top $N$ sentence pairs are selected to train the SMT system. We carry out experiments on 4 different language directions with 9-15M sentence pairs in each direction. The test domains include short message (sms), tweets, and Facebook posts, etc. The experimental results show that our method is able to select a small amount of training data that is used to create a system which outperforms baseline systems trained with all the general-domain data. For example, we obtain over 3.1 BLEU improvement on the Chinese-to-English sms task with around 3% of the whole training data. Experiments also show that we can reduce the size of the in-domain sample to around 100 sentences and still obtain a 2.1 BLEU improvement.

## 2 Related Work

### 2.1 SMT adaptation techniques

Domain adaptation to SMT systems has recently received considerable attention. Based on the availability of in-domain bilingual or monolingual training data, there are several adaptation scenarios. Different domain adaptation techniques, including self-training, data selection, data weighting, etc., have been developed for different scenarios.

Self-training (Ueffing and Ney, 2007; Schwenk, 2008; Bertoldi and Federico, 2009) uses general-domain bilingual parallel data and in-domain monolingual data. An MT system is first trained on bilingual general-domain data, then it is used to translate in-domain monolingual data. The resulting target sentences or bilingual sentence pairs are then used as additional training data for language model or translation model training.

Some early data selection approaches (Zhao et al., 2004; Lü et al., 2007; Moore and Lewis, 2010) use in-domain monolingual data to select monolingual or bilingual data that are similar to the in-domain data according to some criterion. By contrast, (Axelrod et al., 2011; Duh et al., 2013; Axelrod et al., 2015) search for bilingual parallel sentences using the difference in language model perplexity between two language models trained on

in-domain and out-domain data, respectively.

Data weighting approaches weight each data item according to its relevance to the in-domain data. Mixture model adaptation (Foster and Kuhn, 2007; Foster et al., 2010; Sennrich, 2012; Foster et al., 2013) assumes that the general-domain data can be clustered to several sub-corpora, with some parts that are not too far from test domain. It combines sub-models trained on different sub-corpus data sets linearly or log-linearly with different weights. Vector space model adaptation (Chen et al., 2013) has the same assumption, and it weights each phrase pair based on vector space model (VSM). (Chen et al., 2014) improved the VSM adaptation by extending it to distributed VSM and grouped VSM. Instance weighting adopts a rich set of features to compute weights for each instance in the training data; it can be applied to sentence pairs (Matsoukas et al., 2009) or phrase pairs (Foster et al., 2010).

If in-domain comparable data are available, (Daume III and Jagarlamudi, 2011; Irvine et al., 2013) propose mining translations from the comparable data to translate out-of-vocabulary (OOV) words and capture new senses for the new test domains. (Dou and Knight, 2012; Zhang and Zong, 2013) learn bilingual lexical or phrase tables from in-domain monolingual data with a decipherment method, then incorporate them into the SMT system.

All the above approaches assume that either there is an in-domain (mono-lingual, parallel, or comparable) data set with a reasonable size available, or that some sub-corpora are closer to the test domain than others. There is no previous work considering the scenario where only a tiny amount of in-domain data is available: this is the scenario we address in this paper.

### 2.2 CNNs for text classification

In a text classification task, key phrases (or $n$-grams) can help in determining the class of the text, regardless of their locations in the text. For example, the word "desktop" in a sentence may indicate this sentence has computers as its topic; the phrase "not satisfactory" may indicate that the sentiment of the entire sentence is negative. This kind of strong local information about the class of a text can appear in different regions in the input. Convolutional neural networks (CNNs) are useful for text classification because convolutional and pooling layers allow the model to find such local indicators, wherever they are in the text.

Recently, CNNs have shown promising results on many text classification tasks, such as sentiment analysis (Kalchbrenner et al., 2014; Kim, 2014), topic and sentiment classification (Johnson and Zhang, 2015a; Johnson and Zhang, 2015b), paraphrase identification (Yin and Schütze, 2015), entity relation type classification (Zeng et al., 2014; dos Santos et al., 2015), short-text classification (Wang et al., 2015), event extraction and detection (Chen et al., 2015; Nguyen and Grishman, 2015), question understanding and answering (Dong et al., 2015), and box-office prediction based on reviews (Bitvai and Cohn, 2015), etc.

Within the CNN architecture, people also use word embeddings for text classification. (Kalchbrenner et al., 2014) proposes a CNN framework with multiple convolution layers, with latent, dense and low-dimensional word embeddings as inputs. (Kim, 2014) defines a one-layer CNN architecture with comparable performance to (Kalchbrenner et al., 2014). The word embeddings input to the CNN can be pre-trained, and treated as fixed input, or tuned for a specific task. (Johnson and Zhang, 2015b) extends their "one-hot" CNN in (Johnson and Zhang, 2015a) to take region embeddings trained on unlabeled data as CNN input. CNNs that input word embeddings trained on unlabeled data are considered to be semi-supervised CNNs.

## 3 Semi-supervised CNN

A CNN is a feed-forward network consisting of convolutional and pooling layers. Each neuron in the convolutional layer of a CNN processes a segment of input signals, which could be a region in an image or a window of words in a sentence. The convolution layer consists of a set of kernels that compute the dot product between different segments of the input. The kernel associated with the $l$-th segment of the input x computes:

$$\sigma(\mathrm{W} \cdot \mathrm{w}_l(\mathrm{x}) + \mathrm{b}), \qquad (1)$$

where $\mathrm{w}_l(\mathrm{x}) \in \mathrm{R}^q$ is the input window vector that represents the $l$-th segment of data. Weight matrix $\mathrm{W} \in \mathrm{R}^{m \times q}$ and bias vector $\mathrm{b} \in \mathrm{R}^m$ are shared by all the kernels in the same layer, and are learned during the training process.

Because the convolution kernel allows interaction between different parts of the input, it reduces

the requirement to select features by hand. Important features in a sentence are automatically selected with pooling, which is a form of non-linear down-sampling. It takes the maximum or the average value observed in each of the $d$ dimension vectors over different windows. As a result, information from multiple $d$ dimension vectors is kept in a single $d$ dimensional vector. At training time, both the weight vectors and the bias vectors are learned with stochastic gradient ascent.

### 3.1 One-hot CNN

When applying CNN to NLP tasks, the first layer of the network takes word embeddings as input. Word embeddings can be pre-trained using tools such as $word2vec$ (Mikolov et al., 2013) or $GloVe$ (Pennington et al., 2014), in which case a table lookup is enough. Alternatively, these vectors can be learned from scratch as a step in the network training process. When there are enough in-domain data, training in-domain word embeddings is meaningful. However, when the in-domain data are limited, the word embeddings learned from these data are unreliable. In this case, the input sentence x can be represented with one-hot vectors where each vector's length is the vocabulary size, value 1 at index $i$ indicates word $i$ appears in the sentence, and 0 indicates its absence. A CNN with one-hot vector input is called "one-hot CNN" (Johnson and Zhang, 2015a). $w_l(x)$ can be either a concatenation of one-hot vectors, in which the order of concatenation is the same as the word order in the sentence, or it can be a bag-of-word/$n$-gram vector. The bag-of-word (BOW) representation loses word order information but is more robust to data sparsity. In (Johnson and Zhang, 2015a), a CNN whose input being BOW representation is called $bow$-CNN while input with concatenation of vectors is called $seq$-CNN. The window size and stride (distance between the window centers) are meta-parameters. $\sigma$ in Equation 1 is a component-wise non-linear function such as ReLU. Thus, each kernel generates an $m$-dimensional vector where $m$ is the number of weight vectors or neurons. These vectors from all the windows of each sentence are aggregated by the pooling layer, by either component-wise maximum (max pooling) or average (average pooling), then used by the top layer as features for classification.

### 3.2 Semi-supervised CNN

Although the size of the in-domain data is normally small, the unlabeled data from general domains are much larger and easier to obtain. To exploit large amounts of unlabeled data, we adopt a semi-supervised learning framework similar to (Johnson and Zhang, 2015b). It first learns word embedding from unlabeled data, then generates the text segment embedding based on these unsupervised word embeddings. Both the one-hot vectors from the labeled data and the segment embeddings from unlabeled data are combined to train the CNN classifier.

The word embeddings map each word to a real-valued, dense vector (Bengio et al., 2003). Word embeddings are often learned with an unsupervised learning paradigm: each dimension of the continuous word embeddings aims at capturing a latent feature, reflecting certain syntactic and semantic meanings of the word. A widely used approach for generating useful word embeddings was developed by (Mikolov et al., 2013). This method learns the word embeddings such that the likelihood of generating a word based on its contexts (or generating the context of a given word, aka "skip-gram" model) is maximized. It speeds up the training with the hierarchical softmax strategy and a simplified learning objective, which scales very well to very large training corpora. We adopt the skip-gram model, which intuitively learns a classifier that predicts words conditioned on the central word's vector representation. An advantage of such distributed representations is that words that have similar contexts, and therefore similar syntactic and semantic properties, will tend to be near one another in the low-dimensional vector space.

Given the word embeddings trained from unlabeled data, a sentence is represented as a sequence of $d$-dimensional vectors, which is the input to a convolution network that generates feature vectors for each text segment. The segment vectors and one-hot vectors are fed into another convolution layer, which outputs the classification labels. The second network is trained with the labeled in-domain/out-domain data. Therefore, Equation 1 is replaced with:

$$\sigma(\mathbf{W} \cdot \mathbf{w}_l(\mathbf{x}) + \mathbf{V} \cdot \mathbf{u}_l(\mathbf{x}) + \mathbf{b}), \qquad (2)$$

where $\mathbf{w}_l(\mathbf{x})$ is the one-hot vector obtained from segment $l$ in a sentence, and $\mathbf{u}_l(\mathbf{x})$ is the embed-

ding learned from the unlabeled data (general domain training data), applied to the same segment. We train this model with the labeled data. We update the weights W, V, bias b, and the top-layer parameters so that the designated loss function is minimized on the labeled training data.

## 4 Adaptation based on data selection

We use the in-domain data from the translation task as positive samples, and randomly select the same number of sentences from the general domain data as negative samples. We train the CNN model on the positive and negative samples with the one-hot CNN or semi-supervised CNN described previously. The trained CNN model is then used to classify the sentence pairs in the general domain data. The sentence pairs with high in-domain scores are selected to train the machine translation system.

We classify the source sentence and target sentences separately. The CNN model computes two scores for each sentence pair. The sentence pairs are selected based on the source scores, target scores or the sum of source and target scores. Experiments show that selection based on the sum of the source and target scores achieves the best performance. We empirically determine the number of selected in-domain sentences for each MT system based on experimental results on a separate validation set.

When selecting the negative samples, we either randomly sample from the whole data pool, or select from the sentences which have been labeled as negative in the first round classification. Additional experiments show that the results from these two methods are very similar, so we sample the negative samples from the whole general domain for simplicity.

## 5 Experiments

Our goal is to adapt the MT system when only a tiny amount of in-domain data is available. So in our experiments, we did not consider any domain information about the training data, such as the source of each corpus. What we have is a small development set (dev) and one or more test sets (test) which are in the same domain.

### 5.1 Data setting

We carried out experiments in four different data settings. All four have large amounts of bilin-

gual training data: 9-15M sentences. The first two involve translation into English (en) from Chinese (zh) and Arabic (ar), while the last two involve translation from English to Spanish (es) and Chinese. The training data are all publicly available, either from LDC[1] and transcriptions of TED talks[2], where the data are the mixture of newswire, web crawl, UN proceedings and TED talks, etc., or from WMT[3], where the data are the mixture of Europarl, web crawl, news-commentary, and UN proceedings, etc. The dev and test sets are "short messages (sms)" for the first task, which are also available from LDC; "tweets" for the second task; publicly available "Facebook post" for the remaining two tasks. The last three tasks are from social media - an intriguing new area of application for MT - where in-domain parallel training data are seldom publicly available. Table 1 summarizes the statistics of the training, dev, and test data for all the test sets.

### 5.2 Experiment setup

We experiment with two CNN-based data selection strategies:

1. ohcnn: Data selection by supervised one-hot CNN (Section 3.1)

2. sscnn: Data selection by semi-supervised CNN (Section 3.2)

We employ the dev set as in-domain data. All the supervised CNN models are trained with the in-domain dev data as positive examples and an equal number of randomly selected general-domain sentences as negative examples. All the meta-parameters of the CNN are tuned on held-out data; we generate both $bow$-regions and $seq$-regions and input them to the CNN. We set the region size to 5 and stride size to 1. The non-linear function we chose is "ReLU", the number of weight vectors or neurons is 500. The pooling method is component-wise maximum (max pooling). We use the online available CNN toolkit $conText$[4]. To train the general domain word embedding, we used $word2vec$[5]. The size of the vector was set to 300. We also generate word-embedding-based $bow$-regions and $seq$-regions as additional input to the CNN.

---

[1]https://catalog.ldc.upenn.edu/
[2]https://wit3.fbk.eu/
[3]http://statmt.org/wmt15/
[4]http://riejohnson.com/cnn_download.html
[5]https://code.google.com/archive/p/word2vec/

| language | zh2en | ar2en | en2es | en2zh |
|---|---|---|---|---|
| test domain | sms | tweets | facebook | facebook |
| train origin | LDC&TED | LDC&TED | WMT | LDC&TED |
| train size | 12.20M | 8.97M | 15.23M | 12.20M |
| dev size | 6,016 | 1,000 | 800 | 650 |
| test size | 3,282 | 1,500 | 3,378 | 3,343 |

Table 1: Summary of the data. "sms" means "short message". "facebook" means "Facebook post". Data is given as the number of sentence pairs, "M" represents "million". The tasks "zh2en" and "en2zh" use the same training data.

We compared with four baselines for each task. The first baseline SMT system is trained using all general-domain data. The other three systems are trained with data selected with different LM-based data selection methods as same as in (Duh et al., 2013)[6]. The four baselines are:

1. alldata: All general-domain data.

2. ngram: Data selection by 3-gram LMs with Witten-Bell [7] smoothing (Axelrod et al., 2011)

3. rnnlm: Data selection by recurrent neural network LM, with the RNNLM Toolkit (Duh et al., 2013)

4. comblm: Data selection by the combined LM using ngram & rnnlm (equal weight) (Duh et al., 2013).

All systems are trained with a standard phrase-based SMT system with standard settings, i.e., GIZA++ alignment, phrase table Kneser-Ney smoothing, hierarchical reordering models, target side 4-gram language model, and "gigaword" 5-gram language model for systems with English as the target language, etc.

### 5.3 Experimental results

We evaluated the system using BLEU (Papineni et al., 2002) score on the test set. Following (Koehn, 2004), we use the bootstrap resampling test to do significance testing. Table 2 summarizes the results and numbers of the selected sentences for each task. First, we can see that all the data selection methods improved the performance over the baseline "alldata" with much less

---

[6]The code and scripts for the three baselines are available at http://cl.naist.jp/ kevinduh/a/acl2013/

[7]For small amounts of data, Witten-Bell smoothing had performed better than Kneser-Ney smoothing in our experiments

training data (only around 2.5% to 10% of the whole training data). Consistent with (Duh et al., 2013), the three LM based data selection all got improvements, where "rnnlm" obtained better performance than the "ngram" on average. It is not clear that combining the two language model methods ("comblm") yields further improvement. While the one-hot CNN method "ohcnn" obtained similar improvement as the three LM-based methods on average. The semi-supervised CNN (sscnn) achieved the best performance for all the tasks: its improvements over the "alldata" baseline are 3.1, 1.4, 0.7 and 1.4 BLEU score respectively. It beats "ohcnn" by about 0.5 BLEU point on average.

There are two results worth noticing. First, task 1 (zh2en sms task) obtained very high BLEU improvement through data selection. This is because in this task, there is a 120K in-domain subset within the general-domain data. If we train a system on this in-domain data set, we get 25.7 BLEU on the test set. The LM-based methods did not beat this "in-domain data only (indata)" baseline, while the semi-supervised CNN method performed significantly better than this baseline at $p < 0.05$ level. Second, for the other three tasks, there is no in-domain data component in the general-domain data (that we know of). Even in this case, we achieved up to 1.4 BLEU improvement, which demonstrates the effectiveness of our method: it can select highly suitable in-domain sentences, even when the in-domain data is very limited.

In our second experiment, we examine how many labeled samples are needed to train a strong CNN classifier to select the MT training in-domain data. Fixing the number of MT training sentence pairs to 300K that will be selected by the CNN, we reduce the CNN training data from 6,000 down to 100 sentence pairs in steps. The performance of the resulting MT systems for all five data selection

|  | zh2en | | ar2en | | en2es | | en2zh | |
|---|---|---|---|---|---|---|---|---|
|  | #sent | BLEU | #sent | BLEU | #sent | BLEU | #sent | BLEU |
| alldata | 12.2M | 22.9 | 8.9M | 17.6 | 15.2M | 26.8 | 12.2M | 10.0 |
| ngram | 300K | 25.3** | 800K | 18.2** | 1600K | 26.9 | 400K | 10.5* |
| rnnlm | 300K | 25.6** | 800K | 18.4** | 1600K | 27.0 | 400K | 10.5* |
| comblm | 400K | 25.7** | 800K | 18.4** | 1400K | 27.0 | 500K | 10.4* |
| ohcnn | 300K | 25.3** | 700K | 18.2* | 1200K | 27.1* | 400K | 11.0**+ |
| sscnn | 300K | 26.0**+ | 700K | 19.0**++ | 1300K | 27.5**++ | 300K | 11.4**++ |

Table 2: Summary of the results. Data size is given as number of sentence pairs. The number of selected in-domain sentences is determined by the performance on held-out data. "M" represents million, "K" represents thousand. */** means result is significantly better than the "alldata" baseline at $p < 0.05$ or $p < 0.01$ level, respectively. +/++ means result is significantly better than the best LM-based method at $p < 0.05$ or $p < 0.01$ level, respectively.

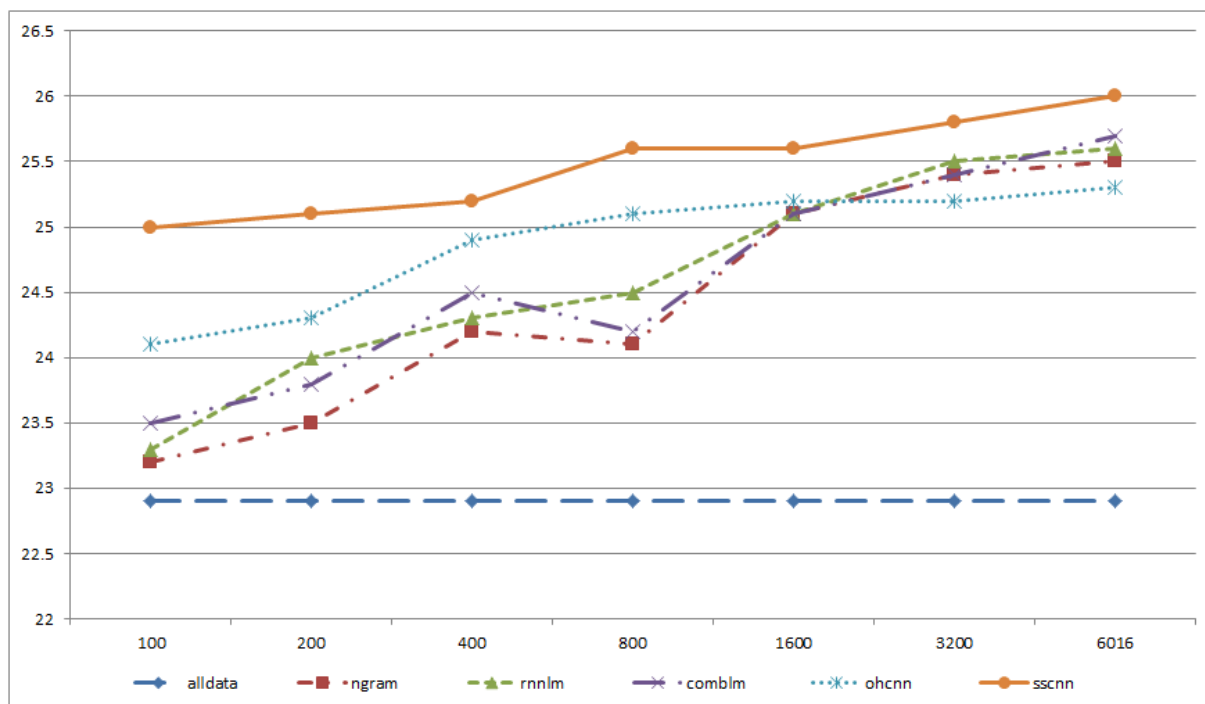

Figure 1: The performance on zh2en sms task with different numbers of in-domain sentences to train LM-based vs. CNN-based classifiers, which are then used to select 300K sentence pairs for MT system training. X-axis is the number of in-domain sentences, Y-axis is BLEU score.

methods is shown in Figure 1.

From Figure 1, we can see that all the data selection methods obtained improvement compared to the "alldata" baseline. When the in-domain training sample is more than 1600 sentence pairs, all the data selection methods obtain reasonable and comparable improvement, while "sscnn" is better than the best LM-based method by 0.3-0.5 BLEU. However, when the in-domain training sample is less than 800 sentence pairs, the difference between the "sscnn" and other methods gets bigger, and CNN-based methods get more stable results than the LM-based methods get. For instance, when the in-domain set increases from 400 to 800, the LM-based methods did not get an improvement; "ngram" and "comblm" even got a small loss on BLEU score. When the in-domain sample is reduced to 100 sentence pairs, the LM-based methods only get a small improvement over the baseline, while the "ohcnn" got a 1.2 BLEU score improvement over the baseline and "sscnn" got a 2.1 BLEU improvement over the baseline. Thus, even if we have no domain knowledge about the training data, when we have only 100 sentences in the test domain, the semi-supervised CNN classifier can still select a good in-domain subset and achieve good performance.

We obtained 2.1 BLEU improvement even when we randomly select only 100 in-domain sentence pairs to train the classification model. Is this just because we luckily sampled a good part of the in-domain data? We repeated the "100 in-domain sentence pairs experiment" three times for our most effective method - "sscnn" - by sampling three different in-domain sets from the whole 6,016-sentence dev set. The average BLEU score we got is 25.03, and the standard deviation is 0.12. This means that our algorithm is quite stable even when the in-domain set is very small.

### 5.4 Discussion

Why do semi-supervised convolutional neural networks perform so well for data selection? We think there are two main reasons. The first one is that convolution captures the important domain information of the words in the window, and the max-pooling operation combines the vectors which, as a result, focuses on the most important "features" in the sentence. Even a highly domain-specific sentence normally contains both domain-specific words and general-domain words. For example, in "*I have a Dell desktop and a Macbook laptop*", the words "*Dell, laptop, Macbook, laptop*" are from the computer domain, while the words "*I, have, a, and*" are general. However, the topic of this sentence is decided by the domain specific words, not the general-domain words. If the properties of the words "*Dell, laptop, Macbook, laptop*" are kept and highlighted, classification will be more accurate for this sentence. The second reason is the use of word embedding learned from the whole general-domain data. A very important advantage of word embedding is that words that have similar meaning will tend to be grouped together in the vector space. If the word "*Lenovo*" in the test sentence is not seen in the labeled data, it would be difficult for LM-based models to classify sentences like "*I prefer choosing a Lenovo machine*" as computer-domain sentence. However, the word embeddings learned from much larger unlabeled data ensure that the word embedding of "*Lenovo*" is close to that of "*Dell*". According to the domain of its neighbor words, the CNN model can still label this sentence as belonging to the computer domain. This property is particularly useful for fast or fine grained adaptation, where obtaining large amount of in-domain samples may be slow or too expensive.

## 6 Conclusions and future work

Domain adaptation with only a tiny amount of in-domain data is a hard problem. In this paper, we proposed to use a semi-supervised convolutional neural network (CNN) to train the domain classification model, then use the CNN to select the data which is most similar to the test domain. Experiments on large data condition SMT tasks showed that this outperforms state-of-the-art language-model-based data selection methods significantly. Particularly when the size of the in-domain data is small, semi-supervised CNN classifier can still select in-domain bilingual sentences to train an adapted SMT system. In future work, we plan to 1) apply this approach to select the data from large size target language corpus, such as the "Gigaword" corpus, for language model training; 2) use the source sentences of the test set to select the data for online dynamic adaptation.

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
