# Peer review of "Semi-supervised Convolutional Networks for Translation Adaptation with Tiny Amount of In-domain Data"

_CoNLL 2016 — decision unknown_

[Official Review · Reviewer 1 · rating 4 · confidence 4]
soundness 4 · originality 3 · clarity 3 · impact 4 · substance 4 · appropriateness 5 · meaningful comparison 4 · replicability 4 · presentation format Poster

The paper describes an MT training data selection approach that scores and
ranks general-domain sentences using a CNN classifier. Comparison to prior work
using continuous or n-gram based language models is well done, even though  it
is not clear of the paper also compared against bilingual data selection (e.g.
sum of difference of cross-entropies).
The motivation to use a CNN instead of an RNN/LSTM was first unclear to me, but
it is a strength of the paper to argue that certain sections of a text/sentence
are more important than others and this is achieved by a CNN. However, the
paper does not experimentally show whether a BOW or SEQ (or the combination of
both( representation is more important and why.
The textual description of the CNN (one-hot or semi-supervised using
pre-trained embeddings) 
is clear, detailed, and points out the important aspects. However, a picture of
the layers showing how inputs are combined would be worth a thousand words.

The paper is overall well written, but some parentheses for citations are not
necessary (\citet vs. \citep) (e.g line 385).

Experiments and evaluation support the claims of the paper, but I am a little
bit concerned about the method of determining the number of selected in-domain
sentences (line 443) based on a separate validation set:
- What validation data is used here? It is also not clear on what data
hyperparameters of the CNN models are chosen. How sensitive are the models to
this?
- Table 2 should really compare scores of different approaches with the same
number of sentences selected. As Figure 1 shows, the approach of the paper
still seems to outperform the baselines in this case. 

Other comments:
- I would be interested in an experiment that compares the technique of the
paper against baselines when more in-domain data is available, not just the
development set.
- The results or discussion section could feature some example sentences
selected by the different methods to support the claims made in section 5.4.
- In regards to the argument of abstracting away from surface forms in 5.4:
Another baseline to compare against could have been the work of Axelrod, 2015,
who replace some words with POS tags to reduce LM data sparsity to see whether
the word2vec embeddings provide an additional advantage over this.
- Using the sum of source and target classification scores is very similar to
source & target Lewis-Moore LM data selection: sum of difference of
cross-entropies. A reference to this work around line 435 would be reasonable.

Finally, I wonder if you could learn weights for the sum of both source &
target classification scores by extending the CNN model to the
bilingual/parallel setting.

[Official Review · Reviewer 2 · rating 4 · confidence 3]
soundness 4 · originality 3 · clarity 4 · impact 3 · substance 4 · appropriateness 5 · meaningful comparison 5 · replicability 3 · presentation format Poster

The paper describes a method for in-domain data selection for SMT with a
convolutional neural network classifier, applying the same framework as Johnson
and Zhang, 2015. The method performs about 0.5 BLEU points better than language
model based data selection, and, unlike the other methods, is robust even if
only a very small in-domain data set is provided. 

The paper claims improvements of 3.1 BLEU points. However, from the results we
see that improvements of this magnitude are only achieved if there are
in-domain data in the training set - training only on the in-domain data
already produces +2.8 BLEU. It might be interesting to also compare this to a
system which interpolates separate in- and out-domain models. 

The more impressive result, in my opinion, comes from the second experiment,
which demonstrates that the CNN classifier is still effective if there is very
little in-domain data. However, the second experiment is only run on the zh2en
task which includes actual in-domain data in the training set, possibly making
selection easier. Would the result also hold for the other tasks, where there
is no in-domain data in the training set? The results for the en2es and en2zh
task already point in this direction, since the development sets only contain a
few hundred sentence pairs. I think the claim would be better supported if
results were reported for all tasks when only 100 sentence pairs are used for
training.  

When translating social media text one often has to face very different
problems from other domains, the most striking being a high OOV rate due to
non-conventional spelling (for Latin scripts, at least). The texts can also
contain special character sequences such as usernames, hashtags or emoticons.
Was there any special preprocessing or filtering step applied to the data?  
Since data selection cannot address the OOV problem, it would be interesting to
know in more detail what kinds of improvements are made through adaptation via
data selection, maybe by providing examples.   

The following remarks concern specific sections:

Section 3.2:
- It could be made clearer how the different vectors (word embeddings, segment
vectors and one-hot vectors) are combined in the model. An illustration of the
architecture would be very helpful. 
- What was the "designated loss function"?

Section 5.2:
For completeness' sake, it could be mentioned how the system weights were
tuned.